# In vivo cell type-specific CRISPR knockdown of dopamine beta hydroxylase reduces locus coeruleus evoked wakefulness

Hiroshi Yamaguchi[1], F. Woodward Hopf[2], Shi-Bin Li[1] & Luis de Lecea [1]

Locus coeruleus (LC) neurons in the brainstem have long been associated with attention and arousal. Optogenetic stimulation of LC-NE neurons induces immediate sleep-to-wake transitions. However, LC neurons also secrete other neurotransmitters in addition to NE. To interrogate the role of NE derived from the LC in regulating wakefulness, we applied in vivo cell type-specific CRISPR/Cas9 technology to disrupt the dopamine beta hydroxylase (*dbh*) gene selectively in adult LC-NE neurons. Unilateral *dbh* gene disruption abolished immediate arousal following optogenetic stimulation of LC. Bilateral LC-specific *dbh* disruption significantly reduced NE concentration in LC projection areas and reduced wake length even in the presence of salient stimuli. These results suggest that NE may be crucial for the awakening effect of LC stimulation and serve as proof-of-principle that CRISPR gene editing in adult neurons can be used to interrogate gene function within genetically-defined neuronal circuitry associated with complex behaviors.

[1] Department of Psychiatry and Behavioral Sciences, Stanford University, 1201 Welch Road, Stanford, CA 94305, USA. [2] Alcohol and Addiction Research Group, Department of Neurology, University of California at San Francisco, 675 Nelson Rising Lane, San Francisco, CA 94158, USA. Correspondence and requests for materials should be addressed to L.d.L. (email: llecea@stanford.edu)

nterrogation of gene function within defined neuronal circuits remains a significant challenge. Conditional gene knockouts require breeding at least two mouse lines with significant time investment and are usually limited to a single locus. RNAi technology allows to reduce expression levels of multiple genes at a time, but lacks cellular specificity and its efficiency depends on endogenous mRNA abundance. Recently developed CRISPR/Cas9-based technology enables investigators to edit genes bypassing the need to engineer germline-modified mutant strains[1,2]. In spite of its very broad range of applications, very few reports have demonstrated efficient CRISPR/Cas9 gene editing in post-mitotic neurons. Here we use Cre-inducible CRISPR/Cas9 mice crossed with Th-IRES-cre knockin mice to generate proof-of-concept evidence that gene editing can be reliably achieved in genetically-defined neuronal ensembles in adult animals.

The locus coeruleus (LC) in the brainstem is the main source of central norepinephrine (NE) and projects to multiple brain areas including olfactory bulb, hippocampus, amygdala and cerebral cortex. LC neurons fire tonically at 1–3 Hz in awake animals, are less active during NREM sleep and almost silent during REM[3]. We previously showed a causal relationship between LC-NE firing, sleep-to-wake transitions, and maintenance of wakefulness using optogenetic tools[4]. However, the LC produces other neurotransmitters in addition to NE[5–7]. For example, the LC secretes dopamine, a precursor of NE, into the hippocampus to enhance the formation of memory[8–10]. Since dopamine also has strong wake-promoting effect[11], it is not yet clear whether the release of NE itself is essential for the arousing effects of LC-NE neurons. Also, animals deficient in the gene of dopamine beta hydroxylase (*dbh*), an enzyme necessary for NE synthesis, have a mild sleep phenotype[12,13], but it is unclear which NE-producing neurons are necessary for control of arousal.

Here we disrupted the *dbh* gene in LC-NE neurons by combining LC-specific Cas9 expression with AAV-mediated sgRNA delivery to interrogate the role of NE from the LC in regulating wakefulness. We show that the LC-specific disruption of *dbh* reduces NE concentration in the LC and projection areas including the prefrontal cortex and prevents immediate arousal following LC photostimulation. In addition, LC-specific *dbh*-disrupted mice fail to show typical arousal responses in the presence of salient stimuli. We, therefore, conclude that NE is crucial to maintain and induce wakefulness.

## Results

**Dbh gene disruption by CRISPR/Cas9.** To disrupt the *dbh* gene in a LC-NE-specific manner, we first crossed the Cre-dependent Cas9 knockin mouse[2] with the noradrenergic-directed driver (Th-IRES-Cre) mice and confirmed expression of Cas9 in the LC-NE of Th-IRES-Cre/Cas9 double heterozygous progenies (Th/Cas9) (Fig. 1a). Also, we confirmed the expression of Cas9 in the VTA dopaminergic neurons in Th/Cas9 as control (Supplementary Fig. 1). We used Th/Cas9 mice for subsequent experiments. For efficient disruption of the *dbh* gene in adult mouse brain, we adopted a dual-target CRISPR strategy in which two different single guide RNAs (sgRNA) target the same gene[14]. To select efficient sgRNAs, we designed four different guide sequences (sgDbh) target the proximal exons of *dbh* (Fig. 1b) and cloned them individually into the Cas9 expressing plasmid (pX458). Based on the ratio of T7E1-digested and -undigested DNA, expression of sgDbh, along with Cas9, induced mutations in the target loci at the percentage of 0, 11, 16 or 9% for sgDbh1, sgDbh2, sgDbh3 or sgDbh4, respectively (Supplementary Fig. 2). We next constructed an AAV vector encoding dual sgDbh (sgDbh3 and sgDbh4) targeting different exons and a cre-dependent fluorescent marker (Fig. 1c) and then packaged it into

the DJ serotype (AAV-DJ sgDbh)[15]. Also, we constructed an AAV vector encoding two non-targeting sgRNAs as control virus (AAV-DJ sgControl).

Injection of AAV-DJ sgDbh into the LC dramatically reduced Dbh immunoreactivity in Th-positive neurons by 85% compared to the contralateral injection of AAV-DJ sgControl (Fig. 1d, e and Supplementary Fig. 3). This penetrance is equivalent to cell-specific mutations induced by injecting cre-expressing AAVs into floxed animals[16,17]. The expression of mCherry, a marker of viral expression, was similar in both sides, strongly indicating that target neurons were transcriptionally intact (Fig.1d, e). Also, the numbers of Th + neurons in the LC were unaltered by sgDbh viral infection, indicating that the expression of sgDbh was not detrimental for the LC-NE neurons (Fig.1e right). To determine the extent of CRISPR/Cas9-introduced mutations in the *dbh* gene, we extracted genomic DNA from LC tissue punches from mice transduced with AAV-DJ sgDbh. Deep sequencing of PCR-amplified targeted exons revealed indel mutations near the predicted Cas9 cleavage site (Fig. 1f, g) and > 80% of the indels led to out-of-frame mutations (Fig. 1f, g).

**Dbh gene disruption does not affect basic properties of LC neurons.** We next examined the electrophysiological properties of LC neurons with cell-attached recordings. As shown in Fig. 2a, firing rate was not altered between sgControl and sgDbh expressing LC cells (Fig. 2a, $n = 7$ sgControl, $n = 8$ sgDbh). Also, the cells linearly responded to ever-increasing hyperpolarization from a voltage near the resting potential (-70 mv). This linear response is characteristic of LC cells[18]. The current responses to hyperpolarizing voltage were not different between sgControl and sgDbh-expressing LC cells (Fig. 2b, $n = 4$ sgControl, $n = 5$ sgDbh). Taken together, these results indicated that *dbh* gene disruption by CRISPR/Cas9 did not affect basic excitability of LC cells. We next determined whether *dbh* gene disruption resulted in reduced production of NE by using an ELISA with extracts from LC and its target areas[19,20]. We found that the amount of NE was significantly decreased in the LC, olfactory bulb and prefrontal cortex in mice injected bilaterally with AAV-DJ sgDbh (71% decrease in LC; $p = 0.002$, 69% decrease in OB; $p = 0.0048$, 50% decrease in PFC; $p = 0.036$) compared to mice injected with AAV-DJ sgControl (Fig. 2c, d). On the other hand, the concentration of dopamine was comparable between sgControl- and sgDbh-infected LC-NE neurons (Fig. 2d right), suggesting that the disruption of *dbh* gene did not accumulate dopamine in the LC.

**NE release is crucial for the awakening effect of LC stimulation.** Our group previously showed that optogenetic stimulation of LC-NE neurons induces immediate sleep-to-wake transitions[4,21]. To test the hypothesis that NE has a major role in the LC-mediated arousal, we injected AAV-DJ sgDbh (left hemisphere) and sgControl (right hemisphere) together with AAV encoding cre-dependent channelrhodopsin-2 (ChR2) into the LC of Th/Cas9 mice (Fig. 3a, b). One week after the injection, we also implanted the mice with optical fibers for subsequent delivery of 473 nm blue light and with electroencephalogram-electromyogram (EEG-EMG) electrodes for simultaneous sleep-wake recordings (Fig. 3a). Phasic 20 Hz optogenetic stimulation of the sgControl-infused LC dramatically reduced the latency of NREM or REM sleep-to-wake transitions and induced long-lasting arousal, as previously described (Fig. 3c-f)[4]. In contrast, photostimulation of sgDbh-infused LC neurons did not significantly alter the latency of NREM or REM sleep-to-wake transitions, relative to control (no light) stimulation trials, indicating that LC-specific depletion of NE blocked the LC-mediated arousal (Fig. 3c-f). In some cases,

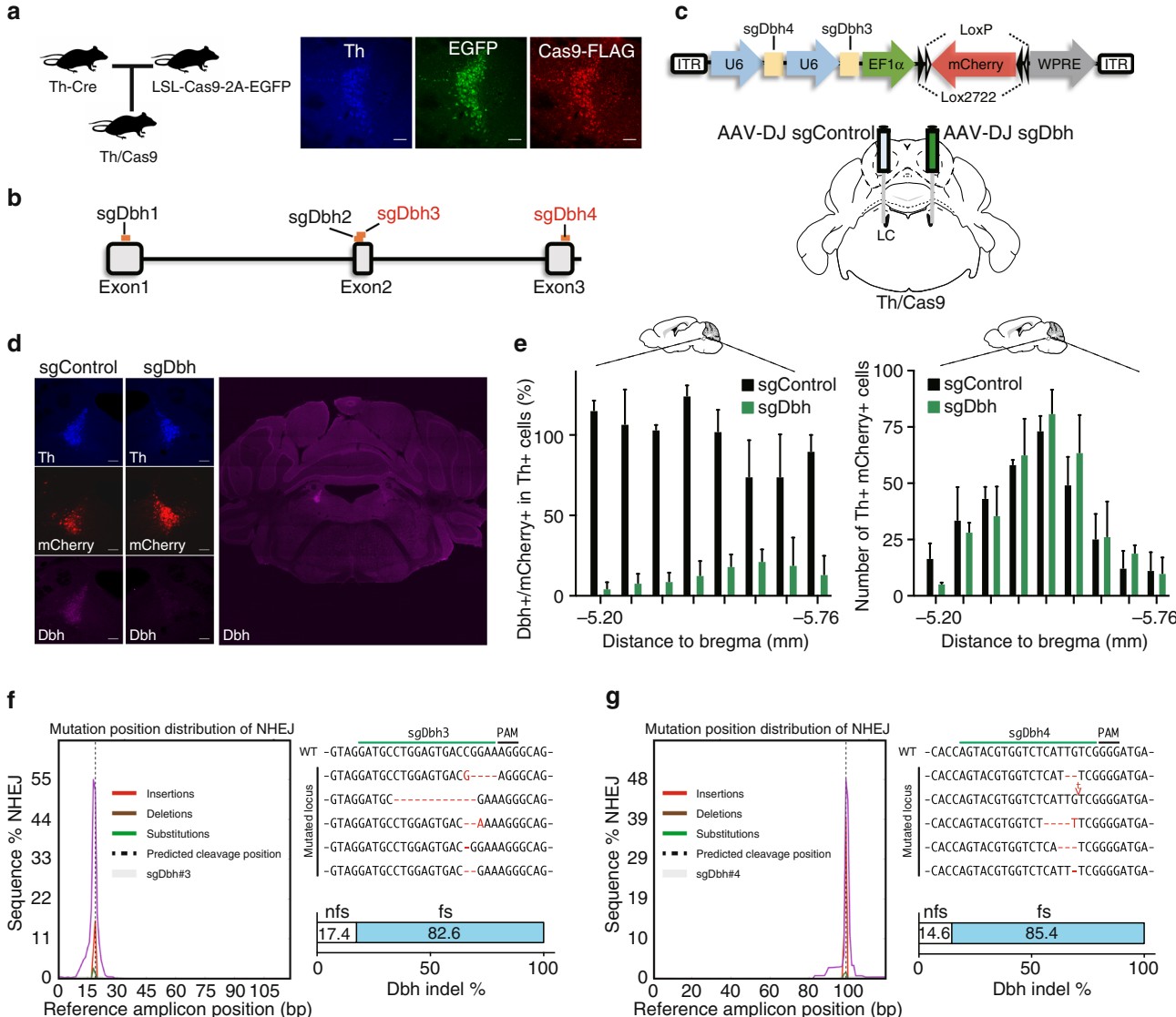

**Fig. 1** Cre-dependent gene editing of *dbh* in the locus coelureus using CRISPR/Cas9. **a** Design for expressing Cas9 in LC-NE (left). Representative immunofluorescence images of Th, EGFP, and Cas9-FLAG in the LC from Th/Cas9 mice (right). **b** sgRNA design for targeting the mouse *dbh* locus. **c** Schematic of the AAV sgDbh vector and experimental design. **d** Representative immunofluorescence images of the LC from Th/Cas9 mice injected with AAV-DJ sgControl (left hemisphere) or AAV-DJ sgDbh (right hemisphere). Scale bar, 100 μm. **e** The percentage of Dbh-positive cells in mCherry- and Th-double positive LC neurons from Th/Cas9 mice injected with AAV-DJ sgDbh or sgControl in 40 μm brain sections from the rostral-to-caudal ends of the LC ($n = 3$ mice; anteroposterior, −5.12 to −5.76). **f, g** *dbh* exon2 (**f**) and exon3 (**g**) mutations analysis of the punch-dissected LC from Th/Cas9 mice injected with AAV-DJ sgDbh. Mutation position distribution of NHEJ (left). Representative mutations (top right). The percentage of frameshift (fs) and non-frameshift (nfs) mutations (bottom right). The LC in **c** and **e** was defined as the Allen Reference Atlas[26]

photostimulation of sgDbh-infused LC neurons induced a brief awakening during 10-sec of stimulation (Fig. 3c, d) but it did not last after the stimulation period. In addition to the raw latency, we also measured the probability of sleep-to-wake transitions following LC photostimulation. Whereas 20 Hz-photostimulation in sgControl-infused LC during NREM or REM sleep elicited an immediate arousal in about 90 or 70% of the trials, respectively, photostimulation in sgDbh-infused LC did not significantly change the probability of sleep-to-wake transitions, relative to no-light control trials (Fig. 3g, h). To exclude the possibility that these effects were due to CRISPR/Cas9-mediated off-target mutagenesis, we constructed another AAV vector encoding different dual sgDbh (sgDbh5 and sgDbh6). Then, we injected AAV-DJ sgDbh 5-6 together with AAV encoding cre-dependent ChR2 into the LC of Th/Cas9 mice. 3 weeks after the virus

injection, we confirmed the disruption of Dbh gene in the LC infected with the virus (Supplementary Fig. 4a). Also, photo-stimulation of re-designed sgDbh-infused LC neurons did not induce immediate sleep-to-wake transitions (Supplementary Fig. 4b, c). Taken together, these results suggest that NE is crucial for LC-mediated arousal.

**NE released from the LC is involved in the maintenance of wakefulness**. To determine the chronic effect of LC-specific NE depletion on spontaneous sleep-wake cycle, we bilaterally injected AAV-DJ sgDbh or AAV-DJ sgControl into the LC of Th/Cas9 mice and implanted EEG-EMG electrodes. We next monitored the duration of wake, NREM, and REM sleep episodes over a 24-h period. Bilateral LC-specific *dbh* gene disruption decreased

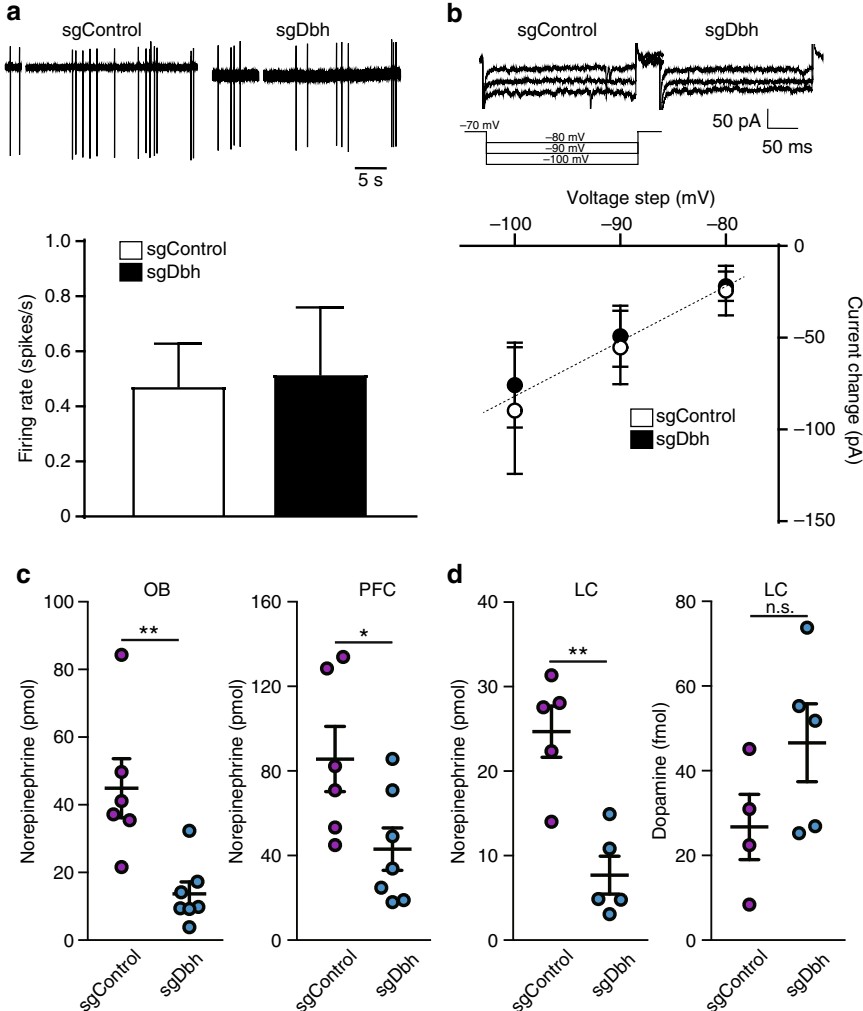

**Fig. 2** Excitability and neurotransmitter productions of *dbh*-disrupted LC neurons. **a**, **b** Firing rate (**a**) and the current clamp recordings (**b**) of the LC cells. The horizontal brain slices containing LC were prepared from the mice bilaterally injected with AAV-DJ sgControl or AAV-DJ sgDbh. The recordings were performed with 7 cells/3 animals (sgControl) and 8 cells/3 animals (sgDbh), respectively. **c** Quantification of norepinephrine in brain homogenates. The olfactory bulb (OB; left panel) or prefrontal cortex (PFC; right panel) were removed from the mice bilaterally injected with AAV-DJ sgControl ($n = 6$) or AAV-DJ sgDbh ($n = 7$). (*$P < 0.05$, **$P < 0.01$ by two-tailed Student *t*-test between sgControl and sgDbh; $t = 3.516$ (OB) and 2.382 (PFC), df = 11 (OB) and 11 (PFC)). **d** Quantification of norepinephrine (left panel) and dopamine (right panel) in the LC. The LC was removed from the mice unilaterally injected with DJ sgControl ($n = 4$–5) or AAV-DJ sgDbh ($n = 5$). (**$P < 0.01$ by two-tailed Student *t*-test between sgControl and sgDbh; $t = 4.512$ (norepinephrine) and 1.601 (dopamine), df = 8 (norepinephrine) and 7 (dopamine))

the total amount of wake and increased NREM during the dark phase compared to control mice, indicating that LC-specific NE depletion disrupted arousal maintenance (Fig. 4a and Supplementary Fig. 5). To further examine the role of NE from the LC when exposed to salient stimuli, we recorded EEG/EMG of sgDbh- or sgControl-injected mice during the inactive (light) phase in the presence of rat bedding, a stressful stimulus for mice. Although all mice were awakened in response to rat odor in the first 10 min of the 1 h trial, sgDbh-injected mice went to sleep again while sgControl-injected mice remained awake (Fig. 4b). Taken together, these results suggest that NE arising from the LC is involved in the maintenance of spontaneous and salient stimuli-induced wakefulness.

## Discussion
In this study, we adopted the dual sgRNA strategy to improve the efficiency of gene disruption. Because sgDbh and Cas9 are constitutively expressed in the AAV-infected LC neurons, Cas9

cleaves the target sequence until it gets mutated by error-prone NHEJ repair. In Fig. 1f and g, we showed the percentages of frameshift mutations in all mutated *dbh* loci were 82.6 and 85.4%, respectively. Thus, the percentage of at least one frameshift mutation in one allele of exon2 or exon3 is $100(1-(17.4/100 \times 14.6/100)) = 97.4\%$. Therefore, the percentage that both alleles have frameshift mutations resulting in the *dbh* disruption would be $100(0.974 \times 0.974) = 94.9\%$. This estimation is consistent with the 85% reduction in dbh immunoreactivity in sgDbh-infected LC (Fig. 1e).

In addition to de novo synthesis by dbh, norepinephrine reuptake at synapses contributes the amount of norepinephrine in synaptic vesicles. Since we disrupted the *dbh* gene specifically in the LC in this study, the NE amount of LC synaptic vesicles are fully dependent on absorption of NE from other norepinephrine neurons at synapses. Here we showed the wake lengths of LC-specific *dbh*-mutated mice were decreased. Also, we found the amounts of norepinephrine in the LC target areas were reduced. These results indicate that the absorption of norepinephrine

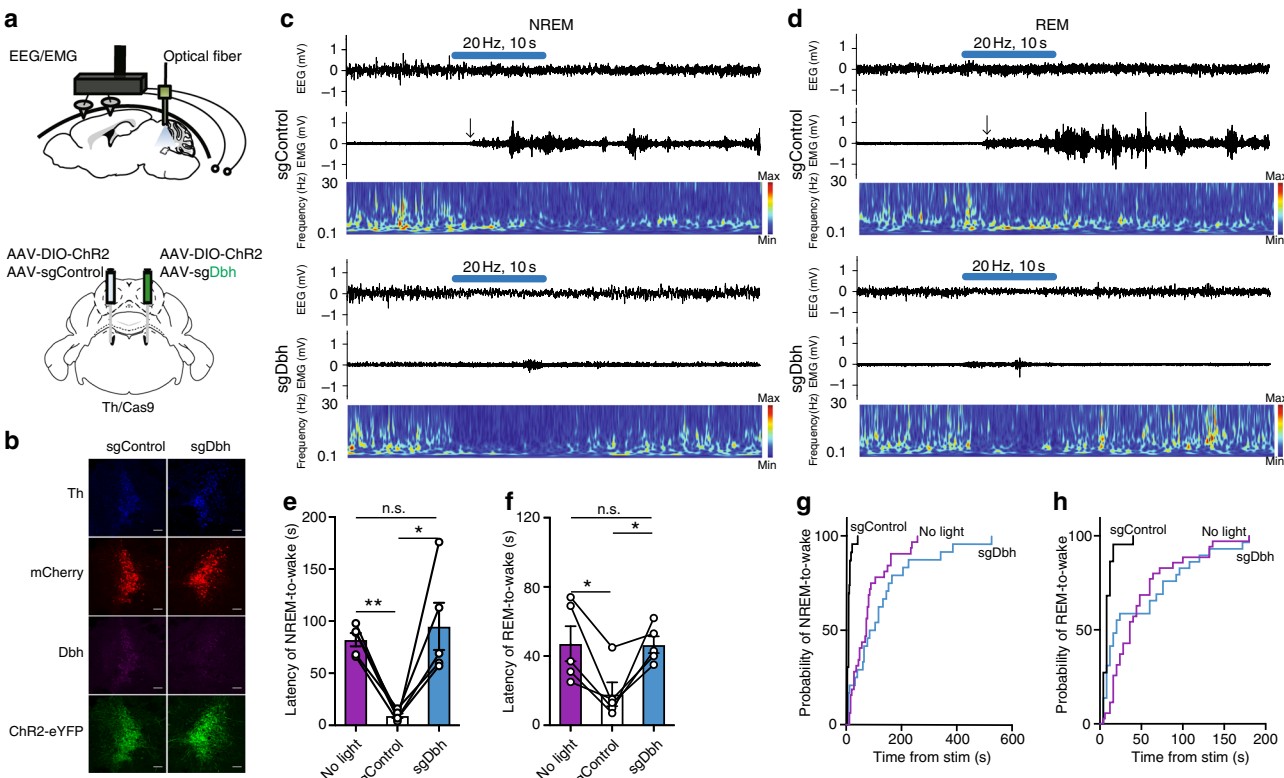

**Fig. 3** Disruption of the *dbh* gene in the LC blocks light-activated ChR2-evoked sleep-to-wake transitions. **a** Schematic of viral injection into the LC, as defined in the Allen Reference Atlas[26], and implantation of an EEG/EMG device and optic fibers. **b** Representative immunofluorescence images of the LC from Th/Cas9 mice injected with AAV-DJ sgControl (left hemisphere) or AAV-DJ sgDbh (right hemisphere) together with AAV-DJ DIO ChR2. Scale bar, 100 μm. **c**, **d** Representative trace of EEG/EMG recordings and the power spectra of EEG (bottom) before and after the unilateral photostimulation (10-ms pulses at 20 Hz for 10 s) of LC neurons in a Th/Cas9 mouse injected with AAV-DJ sgControl (left hemisphere) and AAV-DJ sgDbh (right hemisphere) in NREM (**c**) or REM sleep (**d**). Arrow, the onset of sleep-to-wake transition. **e**, **f** Latencies (mean ± s.e.m) to wake during NREM (**e**) or REM (**f**) sleep following the unilateral photostimulation (10-ms pulses at 20 Hz for 10 s) of LC neurons in a Th/Cas9 mouse injected with AAV-DJ sgControl (left hemisphere) and AAV-DJ sgDbh (right hemisphere) together with AAV-DJ DIO ChR2 ($n = 5$; mean latency of 6–8 stimulations per each mice are plotted) (*$P < 0.05$, **$P < 0.01$; n.s., not significant by one-way repeated measures ANOVA followed by Tukey's multiple comparisons). **g**, **h** Cumulative distribution of sleep-to-wake transitions following the unilateral photostimulation (10-ms pulses at 20 Hz for 10 s) of LC neurons during NREM (**g**) or REM sleep (**h**)

released from other neurons and then absorbed by the LC, was not sufficient for rescuing NE content in the LC.

Drugs that increase extracellular dopamine by inhibiting dopamine transporter promote arousal[22]. Also, we recently found that VTA dopaminergic neurons are involved in the generation and maintenance of wakefulness[11]. As LC neurons secrete both NE and dopamine[8–10], we applied in vivo CRISPR/Cas9 technology to disrupt the *dbh* gene and deplete only NE, not dopamine, from the LC. Unilateral *dbh* gene disruption abolished immediate arousal following optogenetic stimulations of LC. This result is not due to the abnormal excitability of *dbh*$^{-/-}$ LC neurons nor because of increased secretion of dopamine. We confirmed the electrophysiological properties of *dbh*$^{-/-}$ LC neurons were intact (Fig. 2a, b) and these results are consistent with those by Paladini and colleagues using *dbh* knockout mice[18]. Also, we found dopamine does not accumulate in *dbh*$^{-/-}$ LC neurons, possibly because dopamine is also metabolized by other enzymes unaffected by dbh disruption, such as L-monoamine oxidases and catechol-O-methyltransferase. However, it might be possible that activity-dependent release of dopamine is increased as compensation for NE depletion and it might explain the phenotype of LC-specific *dbh* knockout mice. Thus, it would be interesting to examine if type-selective agonists or antagonists of adrenergic receptors could mimic or rescue the phenotype of LC-specific dbh knockout mice.

LC-specific *dbh* disruption blocked immediate awakenings elicited by optogenetic stimulation of the LC, indicating an essential role of NE in the LC to induce arousal. Bilateral LC-specific *dbh* gene disruption further revealed decreased wake and increased NREM sleep amounts during the active phase. These results are consistent with previous studies using *dbh*-null zebrafish[23] and systemic *dbh* knockout mice, which exhibit a decrease in wake and an increase in NREM sleep episode length[12]. In conclusion, our results demonstrate that NE release could be essential for LC-mediated wakefulness. We also show that cell type-specific CRISPR gene editing in adult neurons is an efficient method to interrogate gene function within genetically-defined neuronal circuitry associated with complex behaviors.

## Method

**Animals**. Cre-dependent Cas9 knockin mice[2], a gift from Feng Zhang, were crossed with Tyrosine hydroxylase Cre knockin mice (Th-IRES-Cre; EM:00254), obtained from the European Mouse Mutant Archive. We used double heterozygous mice (Th/Cas9), aged 10–12 weeks at the start of experimental procedures. During all experiments, mice were singly housed in Plexiglas recording chambers at constant temperature (23 ± 1 °C), humidity (40-60%) and circadian cycle (12 h light-dark cycle). Food and water were available *ad libitum*. Mice were randomly assigned to experimental groups, and all groups consisted of age- and sex-matched littermates. All experiments were performed in accordance with the guidelines described in the US National Institutes of Health Guide for the care and Use of Laboratory Animals and approved by Stanford University's Administrative Panel on Laboratory Animal care.

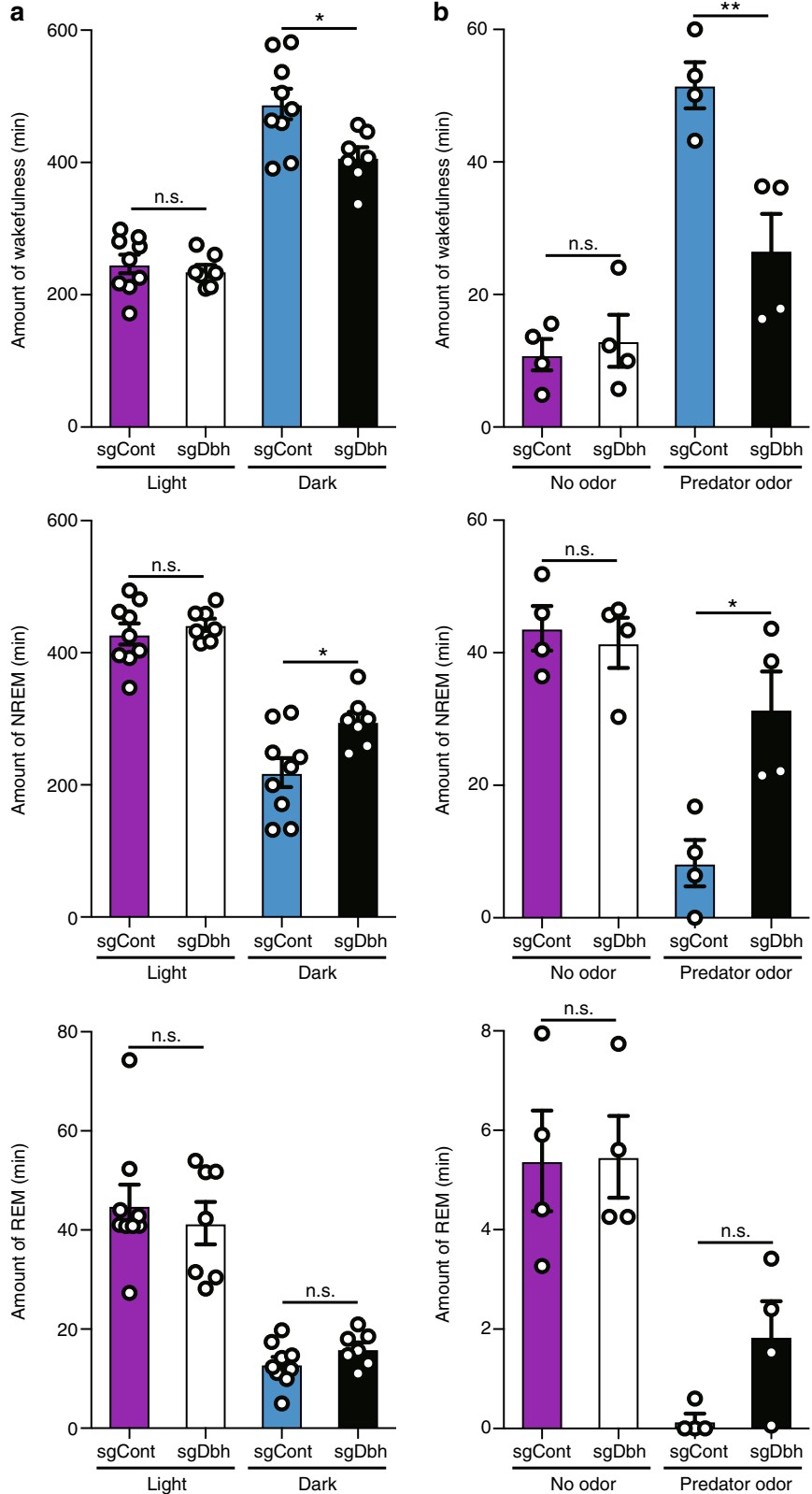

**Fig. 4** Chronic effect of bilateral LC-specific *dbh* disruption on sleep-wake cycle. **a** Total amount of wakefulness (top), NREM (middle), and REM (bottom) sleep during the light or dark phases (*$P < 0.05$; n.s., not significant by two-tailed Student *t*-test between sgControl ($n = 9$) and sgDbh ($n = 7$); $t = 2.73$ (wakefulness) and 2.76 (NREM), d*f* = 14 (wakefulness) and 14 (NREM)). **b** Total amount of wakefulness (top), NREM (middle), and REM (bottom) sleep in the presence of predator odor during the 1-h test period (*$P < 0.05$, **$P < 0.01$; n.s., not significant by two-tailed Student *t*-test between sgControl ($n = 4$) and sgDbh ($n = 4$); $t = 3.814$ (wakefulness), 3.481 (NREM), and 2.342 (REM), d*f* = 6 (wakefulness), 6 (NREM), and 6 (REM))

**DNA constructs and virus**. Dbh target sites for CRISPR/Cas9 were selected by using the ATUM gRNA Design Tool (https://www.atum.bio/eCommerce/cas9/input) and CHOPCHOP (http://chopchop.cbu.uib.no). The target sequences and PAM were as follows; sgDbh1: 5′-CTAAAATCCCTTCGGGGTCCAGG-3′, sgDbh2: 5′-CTGCCCTTTCCGGTCACTCCAGG-3′, sgDbh3: 5′-GATGCCTGGAGTGACCGGAAAGG-3′, sgDbh4: 5′-AGTACGTGGTCTCATTGTCGGGG-3′. sgDbh5: 5′-GGAAGAGCCATTTCAGTCGCTGG-3′, sgDbh6: 5′-TCACAGGGTCCGTTGAACTGGGG-3′. Oligonucleotides encoding guide sequences are purchased from Integrated DNA Technologies (IDT) and cloned individually into BbsI fragment of pX458 (Addgene plasmid 48138[24]). MluI-XbaI flanking U6-sgDbh4 or U6-sgDbh5 and XbaI-MluI flanking U6-sgDbh3 or U6-sgDbh6 sequences were PCR-amplified, respectively using pX458-sgDbh as a template and cloned tandemly into MluI-digested pAAV EF1α DIO mCherry (Addgene plasmid 20299), a gift from Dr. Karl Deisseroth. Non-targeting control guide sequences were also PCR amplified from pX458 empty vector by using same primer pairs and cloned tandemly into pAAV EF1α DIO mCherry in the same way. The primers used were as follows; MluI-F: 5′-GACGACGCGTGAGGGCCTATTTCCCA-3′, XbaI-R: 5′-CTGCTCTAGAAAAAAAGCACCGACTC-3′, XbaI-F: 5′-GACGTCTAGAGAGGGCCTATTTCCCA-3′, MluI-R: 5′-CTGCACGCGTAAAAAAGCACCGACTC-3′. pAAV U6 sgDbh3-U6 sgDbh4 EF1α DIO mCherry, pAAV U6 sgDbh5-U6 sgDbh6 EF1α DIO mCherry and pAAV U6 sgControl-U6 sgControl EF1α DIO mCherry were packaged into AAV-DJ by the gene vector and virus core at Stanford University. The transgene for optogenetic stimulation (AAV-DJ EF1α DIO-ChR2 (H134R)-eYFP) was purchased from the gene vector and virus core at Stanford University.

**Mutational assays with T7 endonuclease I**. CRISPR/Cas9-induced mutations were detected through a mutational assay with T7 endonuclease I. In brief, $2.5 \times 10^4$ NIH3T3 cells were transiently transfected with 0.5 μg of the pX458-sgDbh by using Fugene6 (Promega). 24 h after the transfection, cells were transfected again with the same plasmids. At 48 h post-transfection, the genomic DNA was extracted in 100 microlittle of QuickExtract DNA Extraction solution (Epicentre) and the target locus was amplified by PCR. The primers used were as follows; Dbh Exon1 F: 5′- ATCTTCCTGGTCATCCTGGTG -3′, Dbh Exon1 R: 5′-GAGATCTGCGTTCTCCATCTCT-3′, Dbh Exon2 F: 5′-GGTAGCGGCTGTGATCTCTAAT-3′, Dbh Exon2 R: 5′-GGACTTTAGAATGCAGGGACAG-3′, Dbh Exon3 F: 5′-GATCTTGGAAGAGCCATTTCAG-3′, Dbh Exo3 R: 5′-CGTTTACCATGATGATGTGGTG-3′. 200 ng of PCR products were denatured, re-annealed and digested by 2.5 units of T7 endonuclease I (New England Biolabs). Digested DNA fragments were visualized and quantified by 2% agarose gel electrophoresis.

**Surgery**. Animals were anesthetized by intraperitoneal injection of ketamine (100 mg/kg) and xylazine (10 mg/kg). Recombinant AAV was unilaterally or bilaterally injected into the LC (AP = -5.45 mm; ML = ± 1.05 mm; DV = -3.4 and -4.0 mm) of Th/Cas9 mice. The injection was performed through a 33-gauge needle (Hamilton) attached to a 5.0 μl Hamilton syringe, at a rate of 0.1 μl/min(600 nl total for each two depth). The titers of the virus used were as follows; AAV-DJ U6-sgDbh3 U6-sgDbh4 EF1α DIO mCherry ($4 \times 10^{12}$ gc/ml), AAV-DJ U6-sgControl EF1α DIO mCherry ($5 \times 10^{12}$ gc/ml), AAV-DJ EF1 α DIO ChR2-eYFP ($4 \times 10^{12}$ gc/ml). One week postinjection, mice used for sleep recordings and optogenetic experiments received surgical implantation of custom-made EEG and EMG devices and fiber optic cannulas (200 μm; Doric Lenses, Inc.), above the LC (AP = −5.45 mm; ML = ± 1.05 mm; DV = -3.4). EEG signals were recorded from two miniature screw electrodes on the frontal cortices. EMG signals were recorded from two electrodes inserted in the neck musculature.

**Polysomnographic recording and analysis**. EEG and EMG signals derived from the surgically implanted electrodes were amplified (Grass Technologies) and digitized at 256 Hz using sleep recording software (Vital Recorder, Kissei Comtec America). We digitally filtered and spectrally analyzed the signal by fast Fourier transformation using SleepSign for Animal (Kissei Comtec America). EMG data was filtered with a highpass filter set at 5 Hz to remove the low-frequency artifacts, and then EEG and EMG data were scored according to the criteria as follows: Wakefulness: desynchronized low-amplitude EEG and heightened tonic EMG activity (readout of muscle activities) with phasic bursts; NREM sleep: synchronized, high-amplitude, low-frequency (0.5–4 Hz) irregular pattern EEG and quiescent EMG; REM sleep: EEG pattern dominated by theta rhythm (4–12 Hz) with inactive EMG. We plotted the power spectra of EEG among different groups with Matlab based software package. Experimenters were not blinded to the viral treatment groups during data acquisition.

**Photostimulation and salient stimuli**. Mice were allowed to recover for at least 3 weeks after the surgery and then habituated to a flexible EEG-EMG connection cable and an optical patch cord using zirconia sleeves (Doric Lenses) for 7 days within individual recording chambers. Each cable was flexible so that mice could freely move about their cages. Mice were optically stimulated with blue-light lasers (473 nm, 10 ms pulses at 20 Hz for 10 s) using a waveform generator (Master-8; AMPI) during the light phase (zeitgeber time 5 to 8). We adjusted the light power of the lasers such that the light power exiting the optic fiber was 20 mW.

In experiments with predator odor, we placed freshly prepared rat bedding in mice home cage for a period of 1 h during the light phase (zeitgeber time 6 to 7). EEG-EMG signals were continuously recorded during the test period.

**Illumina sequencing analysis**. Three weeks after viral injection, genomic DNA was extracted from paraformaldehyde-fixed LC tissue punches (AAV-DJ sgDbh-injected; $n = 3$) using QIAamp DNA FFPE Tissue kit (Qiagen). The *dbh* exons were PCR-amplified using the genomic DNA as a template and subjected to library preparation by the Stanford Functional Genomics Facility. The primers used were as follows; Dbh Exon2 NGS F: 5′-CAGACCCTGAGCCTGTCTCT-3′, Dbh Exon2 NGS R: 5′-TAATCCTTGGGGTCACAGGT-3′, Dbh Exon3 NGS F: 5′-AACACCTCAGGCCTGCATAC-3′, Dbh Exon3 NGS R: 5′-CGTGGGGGGTAGCTCAGTG-3′. Illumina's TruSeq indexed pair-ended DNA library preparation protocol was performed automatically on the SPRIworks system (Beckman Coulter, Inc.). After individual libraries were constructed, qualities and band-sizes were assessed using Bioanalyzer High Sensitivity Chip (Agilent Technologies) and Qubit (Life Technologies). Libraries were also quantified by qPCR using the Library Quantification Kit for Illumina sequencing platforms (KAPA Biosystems), using an ABI 7900HT Real-Time PCR System (Life Technologies). Libraries were normalized to a working concentration of 10 nM, using the molarity calculated from qPCR and adjusted for fragment size with the Bioanalyzer analysis. They were finally pooled and sequenced on Illumina's MiSeq. All raw fastq files were extracted for sequencing alignment and further analysis. Illumina sequencing reads were first filtered to remove the ones which have the phred33 quality score under 30. We aligned the remaining reads to reference sequences and determined the mutation positions and the proportion of frameshift versus in-frame coding mutations by using CRISPResso[25].

**Immunohistochemistry**. Mice were anesthetized with ketamine/xylazine and trans-cardinally perfused with 1× PBS, pH 7.4, followed by 4% paraformaldehyde in PBS. Brains were removed, post-fixed for overnight at 4℃, and then cryoprotected in 30% sucrose dissolved in PBS containing 0.1% sodium azide for more than 48 h at 4℃. Brain sectioning was performed on a cryostat (Leica Microsystems) at a thickness of 40 μm. Sections were rinsed in PBS with 0.3% triton X-100 (PBST) and blocked with 2% bovine serum albumin in PBS for 1 h at room temperature. Sections were then incubated in blocking buffer containing primary antibodies for 16 h at room temperature. After 3 × 5 min washes in PBST, sections were incubated in blocking buffer containing secondary antibodies for 2 h at room temperature. After the incubation, sections were washed in PBST three times and then mounted onto MAS-GP microscope slides (Matsunami) and coverslipped with Fluoroshield Mounting Media (Abcam). Images for mCherry, Th, DBH, and FLAG colocalization were collected on a confocal microscope (Zeiss LSM 710, Zen Software). Quantification of colocalization was performed on serial sections from approximate bregma -5.12 to -5.76 from three mice. The antibodies used were as follows; chicken anti-tyrosine hydroxylase (1:2000, AvesLabs #TH), rabbit anti-dopamine beta-hydroxylase (1:500, Immunostar #22806), mouse anti-FLAG M2 (1:1000, Sigma-Aldrich #F1804), Alexa Fluor 405 goat anti-chicken IgY (1:2000, Abcam #ab175674), Alexa Fluor 594 donkey anti-mouse IgG (1:2000, Invitrogen #A21203), Alexa Fluor 647 goat anti-rabbit IgG (1:2000, Invitrogen #A21244).

**ELISA**. A concentration of 20 mg of the olfactory bulb and the prefrontal cortex were dissected out from bilaterally AAV-DJ sgDbh- or AAV-DJ sgControl-injected animals and homogenized in 200 μl of 0.01 N HCl containing 0.15 mM EDTA and 4 mM sodium metabisulfite. The homogenate was centrifuged ($12,000 \times g$, 10 min) and the resulting supernatant was subjected to NE quantification. We determined norepinephrine levels by using an ELISA assay (17-NORHU-E01-RES, ALPCO) according to the manufacturer's instructions. 20 mg of the LC were dissected out from bilaterally AAV-DJ sgDbh- or AAV-DJ sgControl-injected animals and homogenized in 100ul of PBS. The homogenate was centrifuged ($12,000 \times g$, 10 min) and the resulting supernatant was subjected to dopamine quantification. We determined dopamine levels by using an ELISA assay (K-4219, BioVision) according to the manufacturer's instructions.

**Electrophysiology**. Animals were bilaterally injected with AAV-DJ U6-sgDbh3 U6-sgDbh4 EF1· DIO mCherry ($4 \times 10^{12}$ gc/ml), AAV-DJ sgControl into the LC as described above. Three week postinjection, mice were anesthetized with pentobarbital (100 mg/kg), perfused intracardially with a glycerol-based solution (in mM: 252 glycerol, 2.5 KCl, 1.25 NaH$_2$PO$_4$, 1 MgCl$_2$, 2 CaCl$_2$, 25 NaHCO$_3$, 1 L-ascorbate, and 11 glucose), and horizontal brain slices containing LC were cut in the same solution. Slices recovered at ~32 °C in carbogen-bubbled aCSF (containing, in mM: 126 NaCl, 2.4 KCL$_2$, 1.2 NaH$_2$PO$_4$, 1.2 MgCl$_2$, 2.4 CaCl$_2$, 18 NaHCO$_3$, 11 glucose, pH 7.2–7.4, mOsm 302–305) for at least 30 min before experiments, with 1 mM ascorbic acid added just before the first slice. During experiments, slices were submerged and perfused (~2 ml/min) with aCSF. Cells expressing sgDbh were identified by mCherry fluorescence. Action potential firing and responses to voltage steps were recorded using Clampex 10.1 and an Axon 700 A patch amplifier (Molecular Devices, Foster City, CA). A potassium-methanesulfonate based internal solution (in mM: 130 KOH, 105 methanesulfonic acid, 17 HCl, 20 HEPES, 0.2 EGTA, 2.8 NaCl, 2.5 mg/ml Mg-ATP, 0.25 mg/ml

GTP, pH 7.2–7.4, 278–287 mOsm) was used to measure firing in cell-attached mode, and in current-clamp for neurons that broke into whole cell and were then brought to −70 mV by passage of DC current. Responses to voltage steps were measured in voltage clamp by holding cells at −70 mV, then applying 250 msec voltage steps to −80, −90 and −100 mV. Mature adult LC recordings are somewhat technically challenging, and we endeavored to hold cells long enough to record firing and current-voltage responses. In practice, we recorded firing in sgRNA-infected LC cells from 7 sgControl (4 cell attached, 3 current clamp) and 8 sgDbh (4 cell attached, 4 current clamp) LC cells, and hyperpolarizing current steps from 4 sgControl and 5 sgDbh neurons. A lack of effect of dbh knockdown on LC excitability is also observed in dbh KO mice[18].

**Statistics**. Sample sizes were chosen based upon previous publications using optogenetic tools for the study of sleep. Data distribution was assumed to be normal, but not formally tested. We used two-tailed Student $t$-tests for analysis of norepinephrine ELISA (Fig. 2c, d) and the amount of arousal states (Fig. 4a, b). We used RM one-way ANOVA, followed by Tukey's multiple comparisons, for the analysis of latency (Fig. 3e, f). We analyzed all the data using Prism 6.0 (Graphpad Software) and data are presented as mean ± s.e.m. We used Prism 6.0 and Adobe Illustrator CS6 (Adobe Systems) to prepare the figures.

## Data availability

The data sets generated and analyzed as part of this study are available upon request from the corresponding author.

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

## Acknowledgements

We thank Feng Zhang for providing Cre-dependent Cas9 knockin mice. We thank the members of L.d.L's Lab for critical feedback in the preparation of this manuscript. H.Y. was supported by Uehara memorial foundation research fellowship. L.d.L. is supported by National Institutes of Health Grants AG047671, MH087592, MH102638.

## Author contributions

H.Y. and L.d.L. conceived and designed the study. H.Y. performed all experiments and analyzed the data. F.W.H. performed electrophysiological experiments. S.-B.L. analyzed the EEG data. H.Y. and L.d.L. wrote the manuscript.

## Additional information

**Competing interests:** The authors declare no competing interests.

