## [Peer Review File · Nature Communications]

Reviewers' comments:

Reviewer #1 (Remarks to the Author):

In this original study, Yamaguchi & de Lecea used CRISPR/Cas9 technology to knockout dopamine beta hydroxylase (dbh) gene selectively in post-mitotic norepinephrine neurons of the locus neurons (LC-NE) in mice. The authors provide convincing evidence for in vivo targeting of dbh gene and demonstrate the necessity of norepinephrine as a wake- and attention-promoting neuromodulators using in vivo optogenetic stimulation of recombinant dbh KO LC-NE neurons using EEG/EMG sleep recording and behavioural response to rat odors.

The authors provide careful control of CRISPR/Cas9 recombination, control conditions and NE level quantification after LC-NE targeting.

These data are very interesting and provides new tools to manipulate (genetically) neural circuits locally, without the limitations, mainly the time to generate double/triple transgenic line and the temporal resolution of the CRISPR/Cas9 approach.

The manuscript and figures are well presented, experimental procedures and results are convincing, and, supportive of the conclusions.

Although the authors provide convincing evidences that NE alone is supporting the arousal effect of LC-NE neurons, the question remains as to whether LC-NE are expressing/releasing GABA or Glutamate ? Similarly, these data suggest that most of the NE released by LC neurons results from de novo synthesis of NE (through dbh action onto dopamine molecule). Then, what is the role of monoamine vesicular transporter on NE vesicular content. Could the authors discuss this ?

Minor comments:

-L42-46: The link of psychostimulants, wake-promoting effect and Dopamine/Norepinephrine is clear, but the rationale of the parapgraph is unclear. Please reword the sentence.

-few typos

Reviewer #2 (Remarks to the Author):

Yamaguchi and de Lecea applies CRISPR/Cas9 technology in vivo to disrupt the dopamine beta hydroxylase gene in TH+ neurons in the locus coeruleus, and shows a critical role of norepinephrine in the transition to and maintenance of awakening.

The results are straightforward, and interesting both technically and biologically. The manuscript can be accepted conditioned upon adequate addressing of the following questions.

Major comments

- Although the manuscript shows a crucial role of NE, it does not address other neurotransmitters co-released in the LC. One could imagine, in principle, a scenario where

the actual co-release of both DA and NE are instructive for awakening. Therefore, please remove the phrase "and not other neurotransmitters co-released in the LC" in all concluding sentences.

- The results with dbh CRISPR could be interpreted in three alternative ways to the authors' interpretation.

a) Locus coeruleus function may be altered by dbh knockout beyond loss of NE. The authors will need to show intact anatomy, projection, and release properties of LC neurons (release of something other than NE).

b) Presumably, dopamine accumulates as a result of dbh knockout. Could the phenotype due to increased DA release? The authors could address this with a TH knockout which abolishes the synthesis of both DA and NE.

c) The results could be due to off-target effect of the CRISPR approach. The best ways to address this are to have different guide RNAs in different vectors and show the same phenotype, or ideally to perform a rescue experiment with dbh. If a rescue experiment is performed, it would allay some of the concerns in (b) but not (a).

The authors should either address the above concerns experimentally, or discuss them explicitly in the text and change the title to "effect of locus coeruleus dbh", not norepinephrine.

Minor comments

- We have learnt in many scenarios that CRISPR is not efficient in postmitotic cells. It is surprising that the efficiency is so high (85% reduction in DBH immunoreactivity) in this case. Why do the authors think this is the case? Please discuss to highlight the impact of the results.

- Fig 2d, sgControl, EEG. We are not sure what changes we are supposed to see here. Please quantitate and make clear to the readers what about the EEG we should pay attention to.

- In Fig 2d, sgControl, EMG was mislabeled as EEG.

- For Fig 2c and 2d, please display the EMG with different scales – it was almost impossible to see any change with this scale of plotting.

Reviewer #3 (Remarks to the Author):

The MS titled, "In vivo cell type-specific gene editing reveals the awakening effect of locus coeruleus norepinephrine" is about a study showing that a selective disruption of Dbh in LC neurons reduces the arousing effect normally induced by LC activation. This is a straightforward study of broad interest that, on the whole appears to be well carried out.

There are two important controls whose absence may be of concern. First, it is presumed that DA release by these LC TH-expressing cells is unaffected by the manipulation (Dbh mutation) but this has not been tested. If this is beyond the scope of this study then the conclusion, "NE release and not other neurotransmitters co-released in the LC" needs amendment limited to consideration of the necessary role of Dbh.

Second, comparable activation of LC SgDbh to LC control neurons is also presumed. This too should be verified. It is all the more critical in light of figure 2b "typical" labelling of ChR2-eYFP in sgDbh vs sgControl which looks to be different.

Minor points:

1. It might be nice to see what TH/Cas9 looks like in the VTA that should also express TH:Cre.
2. A reference for the DJ serotype would be helpful
3. An earlier study might be referenced for the use of floxed alleles and cre-expressing AAV's (Scammell, et al, JoNS 2003)
4. On line 88 was the "sdDbh or sgControl" meant or should this be an "and" (2 injections in the same mouse).
5. In figure 2g,h what does the probability of firing refer to? There are multiple transitions/mouse and multiple mice (n=?) Are these avg latency probabilities/mouse? If so what is the range of the number of transitions measured/ mouse?
6. In the fig2 legend, rather than "light-evoked" maybe "light-activated ChR-evoked" should be used.
7. It looks like fig2e,f involves matched data- at least for no-light vs sgDbh. This should be illustrated by showing each matched pair and the avg +/-SEM. If all three conditions are matched, please show this as well.

Reviewers' comments:

Reviewer #1 (Remarks to the Author):

In this original study, Yamaguchi & de Lecea used CRISPR/Cas9 technology to knockout dopamine beta hydroxylase (dbh) gene selectively in post-mitotic norepinephrine neurons of the locus neurons (LC-NE) in mice. The authors provide convincing evidence for in vivo targeting of dbh gene and demonstrate the necessity of norepinephrine as a wake- and attention-promoting neuromodulators using in vivo optogenetic stimulation of recombinant dbh KO LC-NE neurons using EEG/EMG sleep recording and behavioural response to rat odors.

The authors provide careful control of CRISPR/Cas9 recombination, control conditions and NE level quantification after LC-NE targeting.

These data are very interesting and provides new tools to manipulate (genetically) neural circuits locally, without the limitations, mainly the time to generate double/triple transgenic line and the temporal resolution of the CRISPR/Cas9 approach.

The manuscript and figures are well presented, experimental procedures and results are convincing, and, supportive of the conclusions.

We do appreciate the time and effort you have dedicated to providing insightful feedback on ways to strengthen our paper.

Although the authors provide convincing evidences that NE alone is supporting the arousal effect of LC-NE neurons, the question remains as to whether LC-NE are expressing/releasing GABA or Glutamate?

While we did not directly examine the release of GABA and Glutamate from the *dbh*^{-/-} LC, we confirmed that firing rate and intrinsic electrophysiological properties were not different between sgControl and sgDbh-expressing LC cells (Figure2a, b). Also, we confirmed the number of LC cells was comparable between *dbh*^{-/-} and control mice (Figure1e). Taken together, these results indicate that expression of sgDbh and Cas9 does not have detrimental effect on LC cells.

Similarly, these data suggest that most of the NE released by LC neurons results from de novo synthesis of NE (through dbh action onto dopamine molecule). Then, what is the role of monoamine vesicular transporter on NE vesicular content. Could the authors discuss this?

Norepinephrine molecules released into synapse are absorbed back into cytoplasm of LC cells by the norepinephrine transporter. Those norepinephrine molecules would be repackaged into synaptic vesicles. This pathway can regulate the amount of norepinephrine in vesicles in addition to *de novo* synthesis. Since we disrupted the *dbh* gene specifically in the LC in this study, the NE amount of LC synaptic vesicles are totally dependent on absorption of NE derived from other norepinephrine neurons. Here, we show the wake lengths of LC-specific *dbh*-mutated mice were significantly reduced. Also, we found the amounts of norepinephrine in the LC target areas are reduced. These results indicate that the amount of norepinephrine, derived from other areas and then absorbed by the LC, was not sufficient to rescue NE content in the LC. We have expanded the discussion to address this issue in main text.

Minor comments:

-L42-46: The link of psychostimulants, wake-promoting effect and Dopamine/Norepinephrine is clear, but the rationale of the paragraph is unclear. Please reword the sentence.

-few typos

We have edited this paragraph to make the rationale clear.

Reviewer #2 (Remarks to the Author):

Yamaguchi and de Lecea applies CRISPR/Cas9 technology *in vivo* to disrupt the dopamine beta hydroxylase gene in TH⁺ neurons in the locus coeruleus, and shows a critical role of norepinephrine in the transition to and maintenance of awakening.

The results are straightforward, and interesting both technically and biologically. The manuscript can be accepted conditioned upon adequate addressing of the following questions.

We wish to express our appreciation to you for their insightful comments on our paper. We feel the comments have helped us significantly improve the paper.

Major comments

- Although the manuscript shows a crucial role of NE, it does not address other neurotransmitters co-released in the LC. One could imagine, in principle, a scenario where the actual co-release of both DA and NE are instructive for awakening. Therefore, please remove the phrase “and not other neurotransmitters co-released in the LC” in all concluding sentences.

As you mentioned, the result of *dbh* knockout alone cannot exclude the possibility that the other neurotransmitters than NE also have the awakening effect. Thus, we deleted the sentences of "and not other neurotransmitters co-released in the LC" in all concluding sentences.

· The results with *dbh* CRISPR could be interpreted in three alternative ways to the authors' interpretation.

a) Locus coeruleus function may be altered by *dbh* knockout beyond loss of NE. The authors will need to show intact anatomy, projection, and release properties of LC neurons (release of something other than NE).

To address this question, we examined the electrophysiological properties of LC neurons with cell-attached recordings. The LC cells were held at -70 mV and their firing rate was determined. Firing rate was not different between sgControl and sgDbh expressing cells (Fig2a). Also, the current responses to hyperpolarizing voltage were not different between sgControl and sgDbh-expressing LC cells (Fig2b). Importantly, these results were consistent with those by Paladini and colleagues (2006), who found no intrinsic differences in LC cells from control and *dbh* KO mice. Also, we confirmed that the numbers of LC cells were comparable between *dbh*^{-/-} and control LC (Fig1e). Taken together, these results indicated that *dbh* disruption with sgRNA and Cas9 does not alter the integrity of LC cells. Thus, we presumed that the phenotype of LC-specific *dbh* knockout mice was not due to the altered function of LC cells beyond loss of NE. These results and discussions were incorporated in main text.

b) Presumably, dopamine accumulates as a result of *dbh* knockout. Could the phenotype due to increased DA release? The authors could address this with a TH knockout which abolishes the synthesis of both DA and NE.

We have added a new Figure.2d to show that the level of dopamine is not altered between *dbh*^{-/-} and control LC neurons. It is probably because dopamine is quickly metabolized by other enzymes than Dbh in the LC cells. We have added this discussion in L161-164.

c) The results could be due to off-target effect of the CRISPR approach. The best ways to address this are to have different guide RNAs in different vectors and show the same phenotype, or ideally to perform a rescue experiment with *dbh*. If a rescue experiment

is performed, it would allay some of the concerns in (b) but not (a).

During the past six months we have tried to rescue the expression of *dbh* in sgDbh-infected LC by injecting AAV carrying sgDbh-resistant *dbh* cDNA to address this question. However, unfortunately, the sgDbh-resistant *dbh* did not express well in the LC. Instead, we showed the number of cells was comparable between *dbh*^{-/-} and control (Fig1e) and also electrophysiological properties of *dbh*^{-/-} LC were intact (Fig2a, b). Therefore, we presumed that the result was not due to the detrimental off-target effect of *dbh* CRISPR.

The authors should either address the above concerns experimentally, or discuss them explicitly in the text and change the title to “effect of locus coeruleus *dbh*”, not norepinephrine.

We have changed the title to “effect of locus coeruleus dopamine *beta*-hydroxylase”.

Minor comments

· We have learnt in many scenarios that CRISPR is not efficient in postmitotic cells. It is surprising that the efficiency is so high (85% reduction in DBH immunoreactivity) in this case. Why do the authors think this is the case? Please discuss to highlight the impact of the results.

In this study, we adopted the dual sgRNA strategy to improve the efficiency of gene disruption. Also, we used AAV-DJ, a new efficient AAV serotype which can infect almost 100% of LC neurons. Because sgDbh and Cas9 are constitutively expressed in the AAV sgDbh infected-LC neurons and then Cas9 cleaves the target sequence until it gets mutated, we can expect all target exons in LC neurons infected with AAV are mutated in either case. In Fig1g and h, we showed the percentage of frameshift mutations in all mutated *dbh* Exon2 and Exon3 after the injection the AAV-sgDbh were 82.6% and 85.4%, respectively. Therefore, focusing on a single allele, the percentage that at least one exon of exon2 or exon3 has a frameshift mutation is $100(1-(17.4/100 \times 14.6/100))=97.4\%$. Thus, the percentage that both alleles have frameshift mutations resulting in the *dbh* disruption would be $100(0.974 \times 0.974) = 94.9\%$. This is the reason why we obtained such a high efficiency of gene disruption *in vivo*. This discussion was added in main text.

· Fig 2d, sgControl, EEG. We are not sure what changes we are supposed to see here. Please quantitate and make clear to the readers what about the EEG we should pay attention to.

- In Fig 2d, sgControl, EMG was mislabeled as EEG.
- For Fig 2c and 2d, please display the EMG with different scales - it was almost impossible to see any change with this scale of plotting.

We have replaced the figure for the better understanding of readers.

Reviewer #3 (Remarks to the Author):

The MS titled, “In vivo cell type-specific gene editing reveals the awakening effect of locus coeruleus norepinephrine” is about a study showing that a selective disruption of *Dbh* in LC neurons reduces the arousing effect normally induced by LC activation. This is a straightforward study of broad interest that, on the whole appears to be well carried out.

There are two important controls whose absence may be of concern. First, it is presumed that DA release by these LC TH-expressing cells is unaffected by the manipulation (*Dbh* mutation) but this has not been tested. If this is beyond the scope of this study then the conclusion, “NE release and not other neurotransmitters co-released in the LC” needs amendment limited to consideration of the necessary role of *Dbh*.

We would like to thank the reviewer for the careful reading and insightful comments on our manuscript. We have added a new Fig2d to show that the level of dopamine is not altered between *dbh*^{-/-} and control LC neurons. Also, we have deleted the phrase “and not other neurotransmitters co-released in the LC” in all concluding sentences to establish a clear focus.

Second, comparable activation of LC Sg*Dbh* to LC control neurons is also presumed. This too should be verified. It is all the more critical in light of figure2b “typical” labelling of ChR2-eYFP in sg*Dbh* vs sgControl which looks to be different.

We have added a new Fig2a, b to show the electrophysiological properties of sg*Dbh*-infected LC neurons are comparable to the control cells. Also, we have replaced the image of ChR2-eYFP to show more “typical” expressions of ChR2-eYFP in the LC.

Minor points:

1. It might be nice to see what TH/Cas9 looks like in the VTA that should also express TH:Cre.

We agree with the reviewer and have added a new Supplemental Fig1 to show that VTA Th-positive neurons also express Cas9.

2. A reference for the DJ serotype would be helpful

We have added a reference of AAV-DJ serotype in L69.

3. An earlier study might be referenced for the use of floxed alleles and cre-expressing AAV's (Scammell, et al, JoNS 2003)

We have added the reference in L74.

4. On line 88 was the "sdDbh or sgControl" meant or should this be an "and" (2 injections in the same mouse).

We have corrected the word.

5. In figure 2g,h what does the probability of firing refer to? There are multiple transitions/mouse and multiple mice (n=?) Are these avg latency probabilities/mouse? If so what is the range of the number of transitions measured/ mouse?

We used 5 mice for each group and 6-8 stimulations per each mouse. That means the data was collected from 30-40 trials. Each plot in a new Fig 3g and h indicates the percentage of trials, resulted in sleep-to-wake transitions, out of all 30-40 trials at any given time after the onset of optogenetic stimulation.

6. In the fig2 legend, rather than "light-evoked" maybe "light-activated ChR-evoked" should be used.

We have corrected the word.

7. It looks like fig2e, f involves matched data- at least for no-light vs sgDbh. This should be illustrated by showing each matched pair and the avg +/-SEM. If all three conditions are matched, please show this as well.

We have corrected the figure to show matched pair.

REVIEWERS' COMMENTS:

Reviewer #1 (Remarks to the Author):

The authors satisfactorily addressed my comments.

Reviewer #2 (Remarks to the Author):

Yamaguchi et al. have revised the manuscript and made significant improvement. We appreciate the extensive efforts of the authors to attempt to rescue the CRISPR knockout effect of dbh, and their change of the title to this manuscript. However, the following points have not been addressed and limit the conclusion. We recommend acceptance of the manuscript, conditioned upon the authors' explicit discussions of the following points to explain these alternative possibilities of interpretation.

- 1) Figure 2D seems underpowered. With enough data points and statistical power, there might actually be a true increase in dopamine.
- 2) The correct projection and release properties of LC neurons (e.g. of dopamine) have not been examined, and if they were not intact, they could explain the phenotype.
- 3) Off-target possibility of CRISPR was not addressed with different guide RNAs or rescue experiments.

Reviewer #3 (Remarks to the Author):

This is a first revision of the manuscript entitled "In vivo cell type-specific gene editing reveals the awakening effect of locus coeruleus dopamine beta-hydroxylase" by Prof de Lecea and colleagues. I believe the authors have successfully resolved all the issues raised by the reviewers for this interesting study, although one point that might benefit from further consideration is the following: Loss of Dbh from LC vesicles may not greatly alter DA levels, however it may increase activity-dependent release of DA in the absence of NE. Since DA may have a higher affinity for Beta-receptors compared to Alpha1-receptors, the overall effect might be a bias for activation of the former at the expense of the latter. Might pharmacologic agents be employed (for example, alpha1 antagonists) to mimic this effect? Or could alpha-1 agonists rescue the KO? This is probably outside the scope of this study, but the possibility might be discussed.

Robert Greene

Reviewer #1 (Remarks to the Author):

The authors satisfactorily addressed my comments.

We do appreciate the time and effort the reviewer have dedicated to improve our paper.

Reviewer #2 (Remarks to the Author):

Yamaguchi et al. have revised the manuscript and made significant improvement. We appreciate the extensive efforts of the authors to attempt to rescue the CRISPR knockout effect of *dbh*, and their change of the title to this manuscript. However, the following points have not been addressed and limit the conclusion. We recommend acceptance of the manuscript, conditioned upon the authors' explicit discussions of the following points to explain these alternative possibilities of interpretation.

1) Figure 2D seems underpowered. With enough data points and statistical power, there might actually be a true increase in dopamine.

We included a discussion regarding the increased release of dopamine and future experiments in discussion part.

2) The correct projection and release properties of LC neurons (e.g. of dopamine) have not been examined, and if they were not intact, they could explain the phenotype.

We did not compare the projection patterns between control and sgDbh-infused LC neurons because it is difficult to precisely quantify the broad projections of LC neurons. Instead, we confirmed the phenotype of LC-specific *dbh* KO mice using re-designed different sgDbh to exclude the off-target detrimental effects of sgDbh on LC neurons.

3) Off-target possibility of CRISPR was not addressed with different guide RNAs or rescue experiments.

To exclude the possibility of off-target effect, we re-designed alternate sgRNAs targeting *dbh* gene. Using the virus carrying these new sgDbh, we confirmed that LC-specific *dbh* disruption blocked immediate arousal following optogenetic stimulation of LC (Supplementary Figure4).

Reviewer #3 (Remarks to the Author):

This is a first revision of the manuscript entitled "In vivo cell type-specific gene editing reveals the awakening effect of locus coeruleus dopamine beta-hydroxylase" by Prof de Lecea and colleagues. I believe the authors have successfully resolved all the issues raised by the reviewers for this interesting study, although one point that might benefit from further consideration is the following: Loss of Dbh from LC vesicles may not greatly alter DA levels, however it may increase activity-dependent release of DA in the absence of NE. Since DA may have a higher affinity for Beta-receptors compared to Alpha1-receptors, the overall effect might be a bias for activation of the former at the expense of the latter. Might pharmacologic agents be employed (for example, alpha1 antagonists) to mimic this effect? Or could alpha-1 agonists rescue the KO? This is probably outside the scope of this study, but the possibility might be discussed.

Robert Greene

We would like to thank the reviewer for insightful suggestions. It will be interesting to see if alpha-1 agonists could rescue the phenotype of the Dbh KO mice in the future. We included this discussion in the manuscript.